# Dual matter-wave inertial sensors in weightlessness

Brynle Barrett[1], Laura Antoni-Micollier[1], Laure Chichet[1], Baptiste Battelier[1], Thomas Lévèque[2], Arnaud Landragin[3] & Philippe Bouyer[1]

Quantum technology based on cold-atom interferometers is showing great promise for fields such as inertial sensing and fundamental physics. However, the finite free-fall time of the atoms limits the precision achievable on Earth, while in space interrogation times of many seconds will lead to unprecedented sensitivity. Here we realize simultaneous $^{87}Rb$–$^{39}K$ interferometers capable of operating in the weightless environment produced during parabolic flight. Large vibration levels ($10^{-2} g\,Hz^{-1/2}$), variations in acceleration ($0$–$1.8\,g$) and rotation rates ($5° s^{-1}$) onboard the aircraft present significant challenges. We demonstrate the capability of our correlated quantum system by measuring the Eötvös parameter with systematic-limited uncertainties of $1.1 \times 10^{-3}$ and $3.0 \times 10^{-4}$ during standard- and microgravity, respectively. This constitutes a fundamental test of the equivalence principle using quantum sensors in a free-falling vehicle. Our results are applicable to inertial navigation, and can be extended to the trajectory of a satellite for future space missions.

[1] LP2N, IOGS, CNRS and Université de Bordeaux, rue François Mitterrand, 33400 Talence, France. [2] CNES, 18 avenue Edouard Belin, 31400 Toulouse, France. [3] LNE-SYRTE, Observatoire de Paris, PSL Research University, CNRS, Sorbonne Universités, UPMC Univ. Paris 06, 61 avenue de l'Observatoire, 75014 Paris, France. Correspondence and requests for materials should be addressed to B.B. (email: brynle.barrett@institutoptique.fr).

The field of quantum physics and atom optics is promising major leaps forward in technology for many applications, including communication, computation, memory and storage, positioning and guidance, geodesy, and tests of fundamental physics. Among these developments, the coherent manipulation of atoms with light, which exploits the particle–wave duality of matter, has led to the development of matter-wave interferometers exhibiting ground-breaking precision[1–4]—particularly for measuring inertial effects such as rotations[5–7] and accelerations[3,4,8,9]. However, the exquisite sensitivity of these quantum inertial sensors often limits their applicability to very quiet and well-controlled laboratory settings—despite recent efforts that have led to major technological simplifications and the emergence of portable devices[3,10,11]. The precision of these instruments becomes particularly relevant when it comes to fundamental tests of general relativity. For instance, the universality of free fall (UFF), a cornerstone of general relativity, which states that a body will undergo an acceleration in a gravitational field that is independent of its internal structure or composition, can be probed at the quantum scale[12,13]. Tests of the UFF generally involve measuring the relative acceleration between two different test masses in free fall with the same gravitational field, and are characterized by the Eötvös parameter

$$\eta = 2\frac{a_1 - a_2}{a_1 + a_2}, \qquad (1)$$

where $a_1$ and $a_2$ are the gravitational accelerations of the two masses. Presently, the most precise measurement of $\eta$ using atom interferometry has been carried out with the two isotopes of rubidium at the level of a few $10^{-8}$ (ref. 14)—five orders of magnitude less precise than the best tests with classical bodies[15,16]. This has motivated increasing the sensitivity of matter-wave interferometers (which scales as the square of the free-fall time) by circumventing the limits set by the gravitational free fall on Earth, either by building a large-scale vertical apparatus[1,17,18] or by letting the entire set-up fall in an evacuated tower[19]. This is also one of the main goals for space-borne experiments[20,21], where the satellite can be viewed as an ideal 'Einstein elevator'.

Our experiment, where two matter-wave sensors composed of rubidium-87 and potassium-39 operate simultaneously in the weightless environment produced by parabolic flight (Fig. 1), represents an atom-interferometric test of the UFF in microgravity. We demonstrate measurements of $\eta$ with precisions of $10^{-3}$ during steady flight and a few $10^{-4}$ in weightlessness using a new interferometer geometry optimized for microgravity operation. Since the aircraft's trajectory during parabolic flight closely mimics that of a satellite in an elliptical orbit, but with residual accelerations of $\sim 1\%$ terrestrial gravity, a precise analysis of its trajectory was necessary to compute the systematic effects on the interferometer phase. This enabled us to quantify the present performance of our atomic sensors onboard the aircraft, and has direct consequences for future implementations of the strap-down inertial navigation algorithm with matter-wave interferometers[10,22,23]. This analysis has also allowed us to put strict requirements on the satellite trajectory in future space missions that target precisions of $\delta\eta \simeq 10^{-15}$ (refs 20,21).

## Results

**Operation during steady flight.** When the aircraft is in steady flight, each of the matter-wave inertial sensors acts as an atom-based gravimeter[3,8,9,24], where counter-propagating light pulses drive Doppler-sensitive single-diffraction Raman transitions between two hyperfine ground states $|1, p\rangle$ and $|2, p + \hbar k^{\mathrm{eff}}\rangle$, where $p$ is the momentum of the atoms resonant with the Raman transition. This creates a superposition of two internal states

separated by the two-photon momentum $\hbar k^{\mathrm{eff}}$, where $\hbar$ is the reduced Planck's constant and $\mathbf{k}^{\mathrm{eff}} \simeq (4\pi/\lambda)\epsilon_z$ is the effective wavevector of the Raman light ($\lambda = 780\,\mathrm{nm}$ for rubidium and $767\,\mathrm{nm}$ for potassium). Because the Raman beams are retro-reflected, this transfer can occur along either the upward ($-\mathbf{k}^{\mathrm{eff}}$) or downward ($+\mathbf{k}^{\mathrm{eff}}$) directions, with an efficiency determined by the vertical velocity $\mathbf{v}$ of the atoms. If the velocity is large enough (for example, the Doppler shift $\mathbf{k}^{\mathrm{eff}} \cdot \mathbf{v}$ is larger than the spectral width $k^{\mathrm{eff}}\sigma_v$ associated with sample's velocity spread $\sigma_v$), a specific momentum transfer direction can be selected by an appropriate choice of the Raman laser frequency difference. Changing the sign of the transfer direction allows the rejection of direction-independent systematics by summing two consecutive, alternated measurements[13,24]. For each transfer direction, the output of the interferometer is given by

$$P^{\pm} = P^0 - \frac{C}{2}\cos(\Phi^{\pm}), \qquad (2)$$

where $P^0$ is the mean probability of finding the atom in one interferometer output port, $C$ is the fringe contrast and $\Phi^{\pm}$ is the total interferometer phase corresponding to a particular momentum transfer direction ($\pm \hbar k^{\mathrm{eff}}$). This phase has contributions from the gravitational acceleration $\phi^{\mathrm{acc}} = \mathbf{k}^{\mathrm{eff}} \cdot \mathbf{a}T^2$ (where $\mathbf{a}$ is the relative acceleration between the reference mirror and the atoms, and $T$ is the free-fall time between light pulses), vibrations of the reference mirror $\phi^{\mathrm{vib}}$, the total laser phase imprinted on the atoms by the Raman beams $\phi^{\mathrm{las}}$, systematic effects $\phi^{\mathrm{sys}}$ and a phase corresponding to a potential violation of the equivalence principle $\phi^{\mathrm{UFF}}_{\mathrm{K,Rb}} = \mathbf{k}^{\mathrm{eff}}_{\mathrm{K,Rb}} \cdot (\mathbf{a}_{\mathrm{K,Rb}} - \mathbf{a})T^2$ for either atomic species.

**Operation during parabolic flight.** To operate in weightlessness, we introduced a new interferometer geometry consisting of two simultaneous single-diffraction Raman transitions in opposite directions, which we refer to as double single diffraction (DSD). In microgravity, the residual Doppler shift is small and the two opposite Raman transitions are degenerate. Thus, we choose a fixed Raman detuning $\delta$ within the spectral width defined by the atomic velocity distribution that simultaneously selects two velocity classes of opposite sign: $\pm \mathbf{v}$. This results in two symmetric interferometers of opposite area (Fig. 2), which sum to yield the output signal for a particular internal state

$$P^{\mathrm{DSD}} = P^+ + P^- = 2P^0 - C\cos(\Sigma\Phi)\cos(\Delta\Phi), \qquad (3)$$

where $2P^0$, $C \leq 1/2$ since the sample is initially split into two velocity classes by the first $\pi/2$-pulse. The DSD interferometer signal given by equation (3) is a product of two cosines—one containing the half-sum $\Sigma\Phi = \frac{1}{2}(\Phi^+ + \Phi^-)$, which exhibits only non-inertial contributions ($\phi^{\mathrm{las}}$, and direction-independent systematics), and one with the half-difference $\Delta\Phi = \frac{1}{2}(\Phi^+ - \Phi^-)$, which contains all inertial contributions ($\phi^{\mathrm{acc}}$, $\phi^{\mathrm{vib}}$ and $\phi^{\mathrm{UFF}}_{\mathrm{K,Rb}}$, and direction-dependent systematics). Since non-inertial and inertial contributions are now separated, we fix $\phi^{\mathrm{las}}$ such that the contrast ($2C\cos(\Sigma\Phi)$) is maximized, and the fringes are scanned by the inertially sensitive phase $\Delta\Phi$. The DSD interferometer has the advantage of simultaneously rejecting direction-independent systematics during each shot of the experiment, since they affect only the fringe contrast. Hence, the systematic phase shift per shot is greatly reduced compared to the single-diffraction configuration.

**Correlated atomic sensor measurements.** Onboard the aircraft the dominant source of interferometer phase noise is caused by vibrations of the reference mirror, which serves as the inertial phase reference for both $^{87}$Rb and $^{39}$K sensors. Hence, the atomic

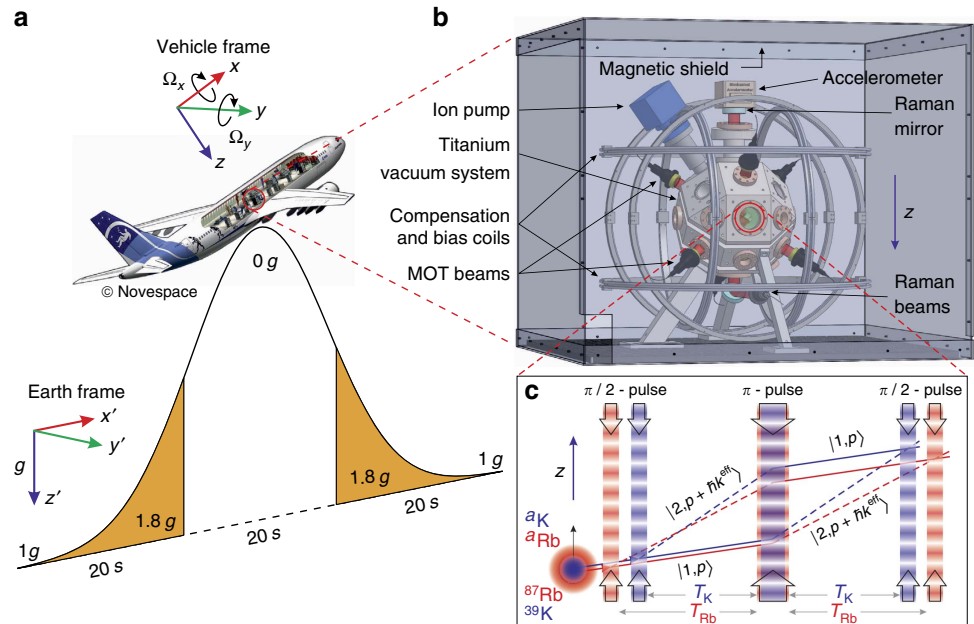

**Figure 1 | Dual matter-wave sensors onboard the Novespace Zero-G aircraft. (a)** Basic trajectory during parabolic flight, which produces 20 s of weightlessness per manoeuvre. The coordinate systems $xyz$ and $x'y'z'$ correspond to the rotating frame of the vehicle and the frame of the Earth, respectively. **(b)** The science chamber mounted onboard the aircraft. Samples of $^{87}$Rb and $^{39}$K are laser-cooled and spatially overlapped in a vapour-loaded magneto-optical trap contained within a titanium vacuum system, which is enclosed by a mu-metal magnetic shield. Raman beams are aligned along the $z$ axis of the aircraft. **(c)** Schematic of the simultaneous dual-species interferometers. Two Mach–Zehnder-type $\pi/2 - \pi - \pi/2$-pulse sequences are centred about the $\pi$-pulse with interrogation times $T_{Rb}$ and $T_K$, respectively. These free-fall times are adjusted independently to equilibrate the scale factors of each interferometer.

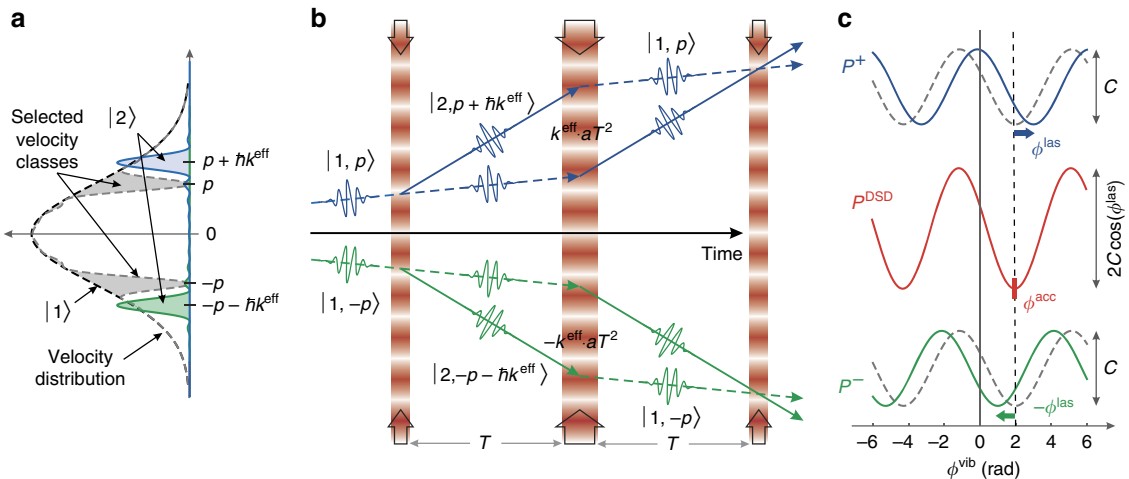

**Figure 2 | Principles of the DSD interferometer. (a)** Velocity distribution of the atoms in microgravity. The Raman frequency is tuned near the half-maximum—simultaneously selecting two symmetric velocity classes with opposite signs. **(b)** DSD interferometer trajectories. Double Raman diffraction[43] is avoided by ensuring that the Rabi frequency is much less than the Doppler frequency difference between the two selected velocities. **(c)** Interference fringes as a function of the vibration phase $\phi^{vib}$ for upward and downward interferometers ($P^{\pm}$) and the sum of the two ($P^{DSD}$). Direction-independent phase shifts like the laser phase $\phi^{las}$ modulate the contrast but not the phase of the DSD fringes (other phase contributions have been omitted for simplicity).

signal caused by its motion is indistinguishable from motion of the atoms. To make this distinction, we measured the mirror motion with a mechanical accelerometer from which we compute the vibration-induced phase $\phi^{vib}$ and correlate it with the normalized output population of each species. We refer to this process as the fringe reconstruction by accelerometer correlation (FRAC) method[10,24,25]. Furthermore, since the two pairs of Raman beams follow the same optical pathway and operate simultaneously, the vibration noise is common mode and can be

highly suppressed from the differential phase between interference fringes.

Figure 3 displays interferometer fringes for both $^{87}$Rb and $^{39}$K, recorded during steady flight ($1\,g$) and in weightlessness ($0\,g$) while undergoing parabolic manoeuvres, for interrogation times $T = 1$ and $2\,ms$. Owing to the large Doppler shift induced by the gravitational acceleration, fringes recorded in $1\,g$ were obtained with the single-diffraction interferometer along the $+\mathbf{k}^{eff}$ direction. Matter-wave interference in $0\,g$ was realized using the

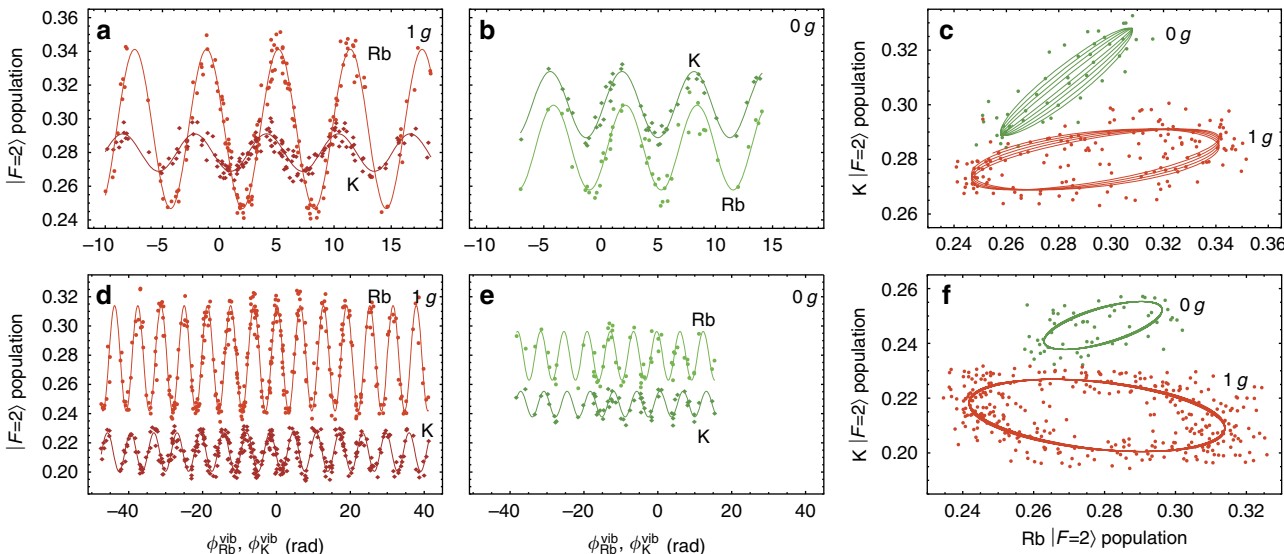

**Figure 3 | Simultaneous K–Rb interferometer fringes during standard- and microgravity.** The normalized population in the ground-state $|F=2\rangle$ for each species is correlated with the vibration-induced phase $\phi^{\text{vib}}$ for an interrogation time $T \simeq 1$ ms (**a–c**) and $T \simeq 2$ ms (**d–f**). Fringes labelled $0g$ were recorded over three consecutive parabolas for **b**, and five parabolas for **e**, consisting of $\sim 12$ points per parabola. Fringes labelled $1g$ were recorded during periods of steady flight between parabolas, and consist of $\sim 70$ points per manoeuvre. In **a,b,d** and **e**, solid lines indicate least-squares fits to sinusoidal functions, which yield a typical SNR of 7 for $^{87}$Rb and 5 for $^{39}$K data. (**c,f**) Correlations between population measurements for each interferometer. The solid lines are parametric representations of the corresponding fit functions shown in **a,b,d** and **e**. The interferometer scale factor ratio was computed using $\kappa = S_K/S_{Rb}$, where $S_j = k_j^{\text{eff}}(T_j + \tau_j^\pi)(T_j + 2\tau_j^\pi/\pi)$ is the exact scale factor for species $j$ (ref. 25). This yielded $\kappa \simeq 0.985$ for **a–c** and $\kappa \simeq 1.001$ for **d–f**. Other parameters: (**a–c**) $T_{Rb} = 1.01$ ms, $T_K = 1$ ms; (**d–f**) $T_{Rb} = 2.01$ ms, $T_K = 2$ ms; (**a–f**) $\pi$-pulse durations: $\tau_{Rb}^\pi = 17 \, \mu$s, $\tau_K^\pi = 9 \, \mu$s.

DSD configuration along both $\pm \mathbf{k}^{\text{eff}}$ simultaneously, which requires a Doppler shift close to zero. Least-squares fits to these fringes yield the FRAC phases $\phi_{Rb,K}^{\text{FRAC}}$, which are related to the gravitational acceleration of each species. From these fits we measure a maximum signal-to-noise ratio of SNR $\simeq 8.9$, and infer an acceleration sensitivity of $(k^{\text{eff}}T^2 \, \text{SNR})^{-1} \simeq 1.8 \times 10^{-4} \, g$ per shot. The best performance onboard the aircraft was achieved with the Rb interferometer at $T = 5$ ms (SNR $\simeq 7.6$), which yielded $3.4 \times 10^{-5} \, g$ per shot—more than 1,600 times below the level of vibration noise during steady flight ($\sim 0.055 \, g$).

Correlation between the potassium and rubidium interferometers is clearly visible when the same data are presented in parametric form (Fig. 3c,f). We obtain general Lissajous figures when the acceleration sensitivity of the two species are not equal[25], as shown in Fig. 3c. These shapes collapse into an ellipse (with an ellipticity determined by the differential phase) only when the interferometer scale factor ratio $\kappa \simeq 1$ (Fig. 3f). This configuration is advantageous because both interferometers respond identically to low-frequency mirror vibrations (that is, frequencies $\lesssim 1/2T$), and the Lissajous shape remains fixed regardless of the common-mode phase span. We achieve this condition by ensuring the interrogation times satisfy $(T_K/T_{Rb})^2 \simeq k_{Rb}^{\text{eff}}/k_K^{\text{eff}}$.

**Tests of the UFF.** Using the sensitivity to gravitational acceleration along the $z$ axis of the aircraft, we made a direct test of the UFF in both standard gravity and in weightlessness. The relative acceleration between potassium and rubidium atoms is measured by correcting the relative FRAC phase shift for systematic effects (see Methods), and isolating the differential phase due to a possible UFF violation

$$\phi_d^{\text{UFF}} = \phi_K^{\text{UFF}} - \kappa \, \phi_{Rb}^{\text{UFF}} = k_K^{\text{eff}} T_K^2 (a_K - a_{Rb}), \quad (4)$$

where $\kappa \simeq k_K^{\text{eff}} T_K^2 / k_{Rb}^{\text{eff}} T_{Rb}^2$ is the ratio of interferometer scale factors when $T$ is much larger than the Raman pulse durations[25].

The Eötvös parameter was then obtained from $\eta = \phi_d^{\text{UFF}}/k_K^{\text{eff}} a^{\text{eff}} T_K^2$, where $a^{\text{eff}}$ is the average projection of the gravitational acceleration vector $\mathbf{a}$ along the $z$ axis over the duration of the measurements. This quantity depends strongly on the trajectory of the aircraft. For our experiments, we estimate $a_{1g}^{\text{eff}} \simeq 9.779(20)$ m s$^{-2}$ and $a_{0g}^{\text{eff}} \simeq 8.56(98)$ m s$^{-2}$ during $1g$ and $0g$, respectively, where the uncertainty is the $1\sigma$ variation of the projection resulting from the aircraft's orientation. We used the Earth gravitational model EGM2008 to estimate changes in local gravity over the range of latitude, longitude and elevation during the flight and found these effects to be negligible compared with those caused by the variation in the aircraft's roll and slope angles. The fact that $a_{0g}^{\text{eff}}$ is less than $g$ originates from the large variation in the aircraft's slope angle over a parabola ($\pm 45°$). From the data shown in Fig. 3d–f, we measure an Eötvös parameter of $\eta_{1g} = (-0.5 \pm 1.1) \times 10^{-3}$ during steady flight. Here the uncertainty is the combined statistical ($\delta\eta_{1g}^{\text{stat}} = 4.9 \times 10^{-5}$) and systematic ($\delta\eta_{1g}^{\text{sys}} = 1.1 \times 10^{-3}$) error—which was limited primarily by direction-independent phase shifts due to the quadratic Zeeman effect. Similarly, in microgravity we measure $\eta_{0g} = (0.9 \pm 3.0) \times 10^{-4}$, with corresponding statistical ($\delta\eta_{0g}^{\text{stat}} = 1.9 \times 10^{-4}$) and systematic ($\delta\eta_{0g}^{\text{sys}} = 2.3 \times 10^{-4}$) errors. Here the increased statistical error is a result of fewer data available in $0g$. However, the systematic uncertainty improves by a factor of $\sim 5$ compared with measurements in standard gravity. This is a direct result of the reduced sensitivity of the DSD interferometer to direction-independent systematic effects. Both measurements are consistent with $\eta = 0$.

**Discussion**

Although the systematic uncertainty was dominated by technical issues related to time-varying magnetic fields, the sensitivity of our measurements was primarily limited by two effects related to the motion of the aircraft—vibrational noise on the retro-reflection mirror and rotations of the interferometer beams.

These effects inhibited access to large interrogation times due to a loss of interference contrast, and are particularly important for future satellite missions targeting high sensitivities with free-fall times of many seconds.

In addition to phase noise on the interferometer, large levels of mirror vibrations cause a loss of interference contrast due to a Doppler shift of the two-photon resonance. To avoid significant losses, the Doppler shift must be well bounded by the spectral width of the Raman transition $\Omega^{\text{eff}}$ during each light pulse. A model of this effect (Supplementary Note 1) confirms that it is most significant when the s.d. of mirror vibrations is $\sigma_a^{\text{vib}} \gtrsim \Omega^{\text{eff}}/k^{\text{eff}}T$. Figure 4a shows the mean power spectral density of vibrations onboard the aircraft during $1\,g$ and $0\,g$. We use these data to estimate upper limits on $T$ corresponding to a relative contrast loss of $\sim 60\%$. For our experimental parameters, we find $T^{\text{max}} \simeq 20$ and $30\,\text{ms}$ for $1\,g$ and $0\,g$, respectively. Conversely, for future space missions planning interrogation times of order $T = 5\,\text{s}$ and $\Omega^{\text{eff}} \simeq 2\pi \times 5\,\text{kHz}$ (ref. 20), our model predicts an upper limit on the vibration noise of $\sigma_a^{\text{vib}} < 40\,\mu g$. One strategy to mitigate this effect is to suppress high-frequency vibrations using an active isolation system modified to operate in microgravity[26]. However, for inertial navigation applications, measuring the vibrations is critical to accurate positioning, thus a hybrid classical-quantum solution may be more viable[27]. Onboard the aircraft, a combination of these two solutions will give access to free-fall times up to $\sim 1\,\text{s}$, above which the jerk of the aircraft will be too large to keep the atoms in the interrogation region defined by the Raman beams.

During parabolic manoeuvres, the aircraft's trajectory is analogous to a Nadir-pointing satellite in an elliptical orbit. The rotation of the experiment during a parabola causes a loss of contrast due to the separation of wave-packet trajectories (Fig. 4c) and the resulting imperfect overlap during the final $\pi/2$-pulse[1,28–30]. For a rotation vector $\mathbf{\Omega}_{\text{T}}$ transverse to $\mathbf{k}^{\text{eff}}$ and a velocity spread $\sigma_v$, the wave-packet displacement can be shown to produce a relative contrast loss of $C \simeq e^{-(k^{\text{eff}}\sigma_v T)^2(|\mathbf{\Omega}_{\text{T}}|T)^2}$ (Supplementary Note 2). Hence, during a parabola where $|\mathbf{\Omega}_{\text{T}}| \simeq 5^\circ\,\text{s}^{-1}$, the loss of contrast reaches 60% by $T = 5\,\text{ms}$ for our [87]Rb sample and by $T = 2.8\,\text{ms}$ for [39]K. Figure 4d,e shows the measured contrast loss as a function of $T$ for each species during both steady and parabolic flight. We fit a model to these data, which includes effects due to both vibrations and rotations. Using only a vertical scale factor as a free parameter, we find good agreement with the data. This loss of contrast can be compensated by counter-rotating the retro-reflection mirror during the interferometer sequence[1,28]. In addition, imaging the atoms on a camera can mitigate this effect, since the rotation-induced spatial fringes in the atomic density profile can be measured directly[1,31]. Using the model we validated with our experiment, we estimate the rotation limitations of a highly elliptical orbit such as in STE-QUEST[20]. In the case of a Nadir-pointing satellite with an orbital rotation rate near perigee (700 km) of $\sim 2.7^\circ\,\text{s}^{-1}$, we estimate a 60% loss of contrast by $T \simeq 73\,\text{ms}$ for the experimental parameters proposed in ref. 20. This justifies the choice of inertial pointing, where the rotation of the satellite counteracts that of the orbit, to reach the target sensitivity of $3 \times 10^{-12}\,\text{m s}^{-2}$ per shot at $T = 5\,\text{s}$. We estimate a loss of $< 1\%$ at $T = 5\,\text{s}$ can be achieved if the residual rotation rate is $< 6 \times 10^{-5\circ}\,\text{s}^{-1}$.

We have realized simultaneous dual matter-wave inertial sensors capable of operating onboard a moving vehicle—enabling us to observe correlated quantum interference between two chemical species in a weightless environment, and to demonstrate a UFF test in microgravity at a precision two orders of magnitude below the level of ambient vibration noise. With the upcoming launch of experiments in the International Space Station[32,33], and in a sounding rocket[34], this work provides another important test bed for future cold-atom experiments in weightlessness. In the Zero-G aircraft, even if the limit set by its motion cannot be overcome, an improvement of more than four orders of magnitude is expected by cooling the samples to ultra-cold temperatures, and actively compensating the vibrations and rotations of the inertial reference mirror. This will approach the desired conditions for next-generation atom interferometry

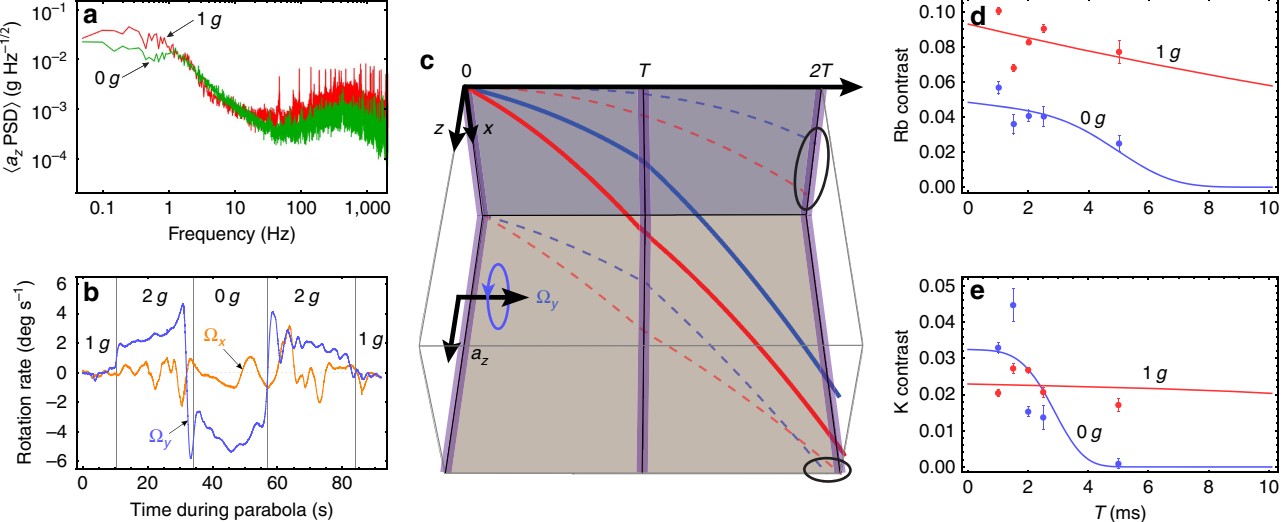

**Figure 4 | Interference contrast loss onboard the aircraft.** (**a**) The mean power spectral density of vibrations along the $z$ axis of the aircraft during $1g$ and $0g$. At low frequencies ($\lesssim 2\,\text{Hz}$), the amplitude of vibrations is approximately five orders of magnitude higher than those found in a quiet laboratory setting[24]. The s.d. of the vibration noise spectrum is $\sigma_a^{\text{vib}} \simeq 0.055g$ during $1g$ and $0.038g$ during $0g$. (**b**) The rotation rates about the $x$ axis (orange line) and $y$ axis (blue line) of the aircraft during a parabola, where $|\Omega_y| \sim 5^\circ\,\text{s}^{-1}$ during parabolic manoeuvres. (**c**) Interferometer trajectories in the presence of a constant acceleration $a_z$ along the $z$ axis and a rotation $\Omega_y$ about the $y$ axis. Purple lines indicate the time of the Raman pulses. The two circled regions in the $xt$ and $zt$ planes show the separation between the two pathways that lead to a phase shift and a loss of contrast. (**d,e**) Measured fringe contrast as a function of $T$ for the Rb and K interferometers during both $1g$ (red points) and $0g$ (blue points). The error bars indicate the statistical uncertainty from least-squares fits to the corresponding fringes. The solid lines are models for the contrast loss for each species (Supplementary Notes 1 and 2).

experiments, such as those designed for advanced tests of gravitation[35], gradiometry[36] or the detection of gravitational waves[37].

## Methods

**Experimental set-up.** Experiments were carried out onboard the Novespace A310 Zero-G aircraft, where the interferometers operated during more than 100 parabolic manoeuvres, each consisting of ~20 s of weightlessness (0 g) and 2–5 min of standard gravity (1 g). Two laser-cooled atomic samples ($^{87}$Rb at 4 μK, and $^{39}$K at 18 μK) were simultaneously interrogated by a $\pi/2 - \pi - \pi/2$ sequence of coherent velocity-sensitive Raman pulses, separated by free-fall times $T_{Rb}$ and $T_K$, respectively (Fig. 1a), which set the acceleration response of each interferometer. The laser light used for this manipulation is aligned through the atoms and retro-reflected along the yaw axis (z axis) of the aircraft (Fig. 1b). A detailed description of our experimental apparatus, fibre-based laser system and fluorescence detection scheme can be found in ref. 35. When the aircraft is in steady flight, the Raman beams are vertical—maximizing the sensitivity to gravitational acceleration. Owing to the high vibration levels onboard the aircraft, the interferometer fringes are reconstructed using a cor-relative method[10,24,25] with measurements from a three-axis mechanical accelerometer (Colibrys SF3600) fixed to the rear of the retro-reflecting mirror. These acceleration measurements were also combined with software to discriminate between the 0 g, 1 g and 2 g phases of a parabola (Fig. 1a), and to automatically switch the interferometers between two different operating modes (single diffraction and DSD) during each manoeuvre. A frequency chirp is applied to the Raman frequency during 1 g to cancel the gravity-induced Doppler shift. The chirp is disabled by software during parabolic manoeuvres. Interferometer measurements taken during the 2 g phase were rejected during the data analysis process. Finally, the rotation rates $\Omega_x(t)$ and $\Omega_y(t)$ are continuously monitored during the flight using a two-axis fibre-optic gyroscope (KVH DSP-1750). Combined with continuous acceleration measurements, we integrate the equations of motion in the rotating frame to obtain the trajectory of the two atomic clouds with respect to the reference mirror for each shot of the experiment. These trajectories are used to estimate systematic shifts on the measurement of $\eta$ due to the Coriolis effect and the magnetic gradient.

**Evaluation of systematic effects.** To evaluate the systematic effects on the measurement of $\eta$, we begin by separating the total interferometer phase $\Phi_j^{\pm}$ (corresponding to atom $j$ and momentum transfer direction $\pm k_j^{eff}$) into five contributions

$$\Phi_j^{\pm} = \pm \phi_j^{acc} \pm \phi_j^{UFF} \pm \phi_j^{vib} + \phi_j^{las} + \phi_j^{sys, \pm}, \quad (5)$$

where $\phi_j^{acc} = \mathbf{S}_j \cdot \mathbf{a}$ is the phase due to the relative gravitational acceleration $\mathbf{a}$ between the reference mirror and the atoms with scale factor $\mathbf{S}_j = \mathbf{k}_j^{eff}(T_j + \tau_j^{\pi})(T_j + 2\tau_j^{\pi}/\pi)$ and $\pi$-pulse duration $\tau_j^{\pi}$, $\phi_j^{UFF} = \mathbf{S}_j \cdot (\mathbf{a}_j - \mathbf{a})$ is a phase shift from a possible UFF violation, $\phi_j^{vib} = \mathbf{k}_j^{eff} \cdot \int f_j(t) \mathbf{a}^{vib}(t) dt$ is a random phase caused by mirror vibrations with corresponding time-dependent acceleration $\mathbf{a}^{vib}(t)$ and interferometer response function $f_j(t)$[25,38], $\phi_j^{las} = \varphi_j(0) - 2\varphi_j(T_j) + \varphi_j(2T_j)$ is the contribution from the Raman laser phase $\varphi_j(t)$ at each interferometer pulse and $\phi_j^{sys, \pm}$ represents the total systematic phase shift. We express the total systematic phase as the following sum

$$\phi_j^{sys, \pm} = \sum_i \phi_{i,j}^{sys, \pm} = \sum_i \Sigma\phi_{i,j}^{ind} \pm \Delta\phi_{i,j}^{dep}, \quad (6)$$

where $i$ is an index corresponding to a given systematic effect. In general, these phases can depend on both the magnitude and the sign of $\mathbf{k}_j^{eff}$. To simplify the analysis, we divide $\phi_{i,j}^{sys, \pm}$ into two separate phases labelled $\Sigma\phi_{i,j}^{ind}$ for the direction-

independent phase shifts and $\Delta\phi_{i,j}^{dep}$ to denote the direction-dependent shifts (that is, those proportional to the sign of $\mathbf{k}_j^{eff}$). We isolate these components by evaluating the sum and the difference between systematics corresponding to each momentum transfer direction

$$\Sigma\phi_{i,j}^{ind} = \frac{1}{2}\left(\phi_{i,j}^{sys, +} + \phi_{i,j}^{sys, -}\right), \quad (7)$$

$$\Delta\phi_{i,j}^{dep} = \frac{1}{2}\left(\phi_{i,j}^{sys, +} - \phi_{i,j}^{sys, -}\right), \quad (8)$$

For the specific case of the single-diffraction interferometer used in 1 g along $+\mathbf{k}_j^{eff}$, the systematic phase shift is given by $\phi_j^{sys, 1g} = \phi_j^{sys, +}$. In comparison, for the DSD interferometer, only direction-dependent systematic effects can shift the phase of the fringes measured as a function of $\phi_j^{vib}$. In the ideal case, the sum of $\Delta\phi_{i,j}^{dep}$ is the sole contribution to the systematic shift of the DSD fringes, since $\Sigma\phi_{i,j}^{ind}$ is direction-independent and thus contributes only to the fringe contrast (Fig. 2). However, in the more general case, these two phases can indirectly affect the phase of the DSD interferometer when the two pairs of Raman beams do not excite the selected velocity classes $\pm v_j^{sel}$ with the same probability. We denote this contribution $\phi_j^{DSD}$, thus the total systematic phase for the interferometers used in 0 g is

$$\phi_j^{sys, 0g} = \phi_j^{DSD} + \sum_i \Delta\phi_{i,j}^{dep}. \quad (9)$$

Table 1 displays a list of the systematic phase shifts affecting the interferometers operated at $T \simeq 2$ ms onboard the aircraft (Fig. 3d–f).

**Phase corrections and $\eta$ measurements.** The raw interferometer phase for each species is measured directly from fits to the fringes reconstructed using the FRAC method (Fig. 3). We refer to this quantity as the FRAC phase $\phi_j^{FRAC}$. For the interferometers used in standard gravity, the measured fringes follow equation (2) with total phase $\Phi_j^+$. Since the vibration phase $\phi_j^{vib}$ is the quantity used to scan $\Phi_j^+$, the FRAC phase is related to the sum of all other phase contributions through

$$\phi_j^{acc} + \phi_j^{las} + \phi_j^{sys, +} + \phi_j^{UFF} = \phi_j^{FRAC} + 2\pi n_j^{2\pi}, \quad (10)$$

where $n_j^{2\pi}$ is an integer representing a certain fringe. Assuming that $\left|\phi_j^{UFF}\right| < \pi$, and provided the total uncertainty from all other phases is much less than $\pi$, the UFF phase can be isolated from equation (10) by computing the fringe number from $n_j^{2\pi} = [(\phi_j^{acc} + \phi_j^{las} + \phi_j^{sys, +})/2\pi]$, where the square brackets indicate rounding to the nearest integer. A similar procedure can be carried out for the DSD fringes obtained in weightlessness, where the total phase $\Phi_j^+$ is replaced with the half-difference $\Delta\Phi_j = \frac{1}{2}(\Phi_j^+ - \Phi_j^-)$. We point out that the laser phase does not contribute to the DSD interferometer because it is independent of the momentum transfer direction. Furthermore, since we are interested in only the differential UFF phase given by equation (4), the contribution due to the gravitational acceleration cancels $(\phi_K^{acc} - \kappa\phi_{Rb}^{acc} = 0)$.

The Eötvös parameter is obtained from

$$\eta = \frac{a_K - a_{Rb}}{a^{eff}} = \frac{\phi_d^{UFF}}{S_K a^{eff}}, \quad (11)$$

where $a^{eff}$ is the effective gravitational acceleration to which the atom interferometer is sensitive over the duration of a measurement. To estimate $a^{eff}$, we first compute the gravitational acceleration along the vertical $z'$ axis, $a(\varphi, \lambda, h)\epsilon_{z'}$, over a two-dimensional grid of latitude ($\varphi$) and longitude ($\lambda$)

---

**Table 1 | Table of systematic phase shifts for the single-diffraction and DSD interferometers.**

| Systematic effect | $+ k^{eff}$ in 1g | | $\pm k^{eff}$ in 0g | | | | Unit |
|---|---|---|---|---|---|---|---|
| | $\phi_{Rb}^{sys}$ | $\phi_K^{sys}$ | $\Delta\phi_{Rb}^{dep}$ | $\Sigma\phi_{Rb}^{ind}$ | $\Delta\phi_K^{dep}$ | $\Sigma\phi_K^{ind}$ | |
| Quadratic Zeeman | 2,127 (48) | 30,596 (694) | 0 | 2,127 (34) | 0 | 30,587 (491) | mrad |
| Magnetic gradient | 31.9 (8.3) | 958 (215) | 20.7 (4.1) | 0.0096 (19) | 745 (116) | 1.46 (21) | mrad |
| Coriolis effect | − 0.551 (18) | − 0.80 (26) | 10.9 (1.5) | 0 | 14.6 (2.1) | 0 | mrad |
| One-photon light shift | − 2.1 (3.6) | − 51 (81) | 0 | − 2.1 (2.5) | 0 | − 51 (57) | mrad |
| Two-photon light shift | 1.3 (2.8) | 16 (68) | 1.3 (2.0) | 0 | 16 (48) | 0 | mrad |
| Extra laser lines | − 0.18 (10) | 0 | 0.030 (26) | 0.19 (16) | 0 | 0 | mrad |
| FRAC method | 0.0 (3.3) | 0.0 (3.3) | 0.0 (3.3) | 0 | 0.0 (3.3) | 0 | mrad |
| Gravity gradient | 22 (20)E-6 | 61 (20)E-6 | 54 (52)E-9 | 15 (14)E-6 | − 20 (8)E-9 | 39 (14)E-6 | mrad |
| DSD asymmetry | 0 | 0 | − 39 (3) | 0 | − 29 (30) | 0 | mrad |
| Total | 2,157 (49) | 31,519 (735) | − 6.8 (6.6) | 2,125 (34) | 795 (129) | 30,537 (494) | mrad |

The 1σ statistical uncertainties are indicated in parentheses. The corresponding interference fringes are shown in Fig. 3d,e.

**Table 2 | Table of phase corrections and final measurements of $\eta$.**

| | $+\mathbf{k}^{\mathrm{eff}}$ in 1$g$ | | $\pm\mathbf{k}^{\mathrm{eff}}$ in 0$g$ | | Unit |
|---|---|---|---|---|---|
| | **Rb** | **K** | **Rb** | **K** | |
| $\phi_j^{\mathrm{acc}}$ | 646.442 | 647.146 | $-$ 0.356 | $-$ 0.356 | rad |
| $\phi_x^{\mathrm{las}}$ | $-$ 646.905 | $-$ 647.132 | 0 | 0 | rad |
| $\phi_j^{\mathrm{sys}}$ | 2.157 (49) | 31.519 (735) | $-$ 0.0068 (66) | 0.795 (129) | rad |
| Sum | 1.694 (49) | 31.532 (735) | $-$ 0.363 (66) | 0.439 (129) | rad |
| $\phi_j^{\mathrm{FRAC}}$ | 3.294 (18) | 1.363 (26) | 2.855 (72) | 3.703 (81) | rad |
| $n_j^{2\pi}$ | 0 | 5 | $-1$ | $-1$ | |
| $\phi_j^{\mathrm{UFF}}$ | 1.597 (52) | 1.246 (735) | $-$ 3.065 (73) | $-$ 3.020 (152) | rad |
| | **K $-\kappa$Rb** | | **K $-\kappa$Rb** | | |
| $\phi_d^{\mathrm{UFF}}$ | $-$ 0.352 (737) | | 0.049 (169) | | rad |
| $\eta$ | $-$ 0.5(1.1) $\times 10^{-3}$ | | 0.9(3.0) $\times 10^{-4}$ | | |

Phase corrections and measurements of $\eta$ are given for the single-diffraction interferometer in 1$g$ and the DSD interferometer in 0$g$. In both cases, $T_K = 2$ ms, $T_{Rb} = 2.01$ ms, $\tau_{Rb}^\pi = 17\,\mu$s and $\tau_K^\pi = 9\,\mu$s—yielding scale factors $S_{Rb} = 65.97$ rad s$^2$ m$^{-1}$ and $S_K = 66.04$ rad s$^2$ m$^{-1}$, and a ratio of $\kappa = 1.0011$. Values of $a^{\mathrm{eff}}$ for both 1$g$ and 0$g$ are given in Table 3. The corresponding data are shown in Fig. 3d,e.

**Table 3 | Inertial parameters measured during each flight configuration.**

| | Steady flight | | Parabolic flight | | Unit |
|---|---|---|---|---|---|
| | **Mean** | **Range** | **Mean** | **Range** | |
| $h$ | 6.332 | 0.025 | 8.642 | 0.228 | km |
| $s$ | 163 | 8 | 82 | 13 | m s$^{-1}$ |
| $a_x$ | $-$ 0.196 | 0.314 | 0.078 | 0.069 | m s$^{-2}$ |
| $a_y$ | 0.078 | 0.039 | 0.039 | 0.039 | m s$^{-2}$ |
| $a_z$ | 9.816 | 0.382 | 0.098 | 0.226 | m s$^{-2}$ |
| $\theta_x$ | $-$ 1.2 | 2.5 | $-$ 1.9 | 2.4 | $^\circ$ |
| $\theta_y$ | 0.01 | 0.35 | $-$ 6.0 | 50.8 | $^\circ$ |
| $\Omega_x$ | $-$ 0.07 | 0.24 | $-$ 0.19 | 0.90 | $^\circ$ s$^{-1}$ |
| $\Omega_y$ | 0.00 | 0.15 | 4.1 | 1.1 | $^\circ$ s$^{-1}$ |
| $\Omega_z$ | $-$ 0.04 | 0.12 | 0.00 | 0.16 | $^\circ$ s$^{-1}$ |
| $\langle a\rangle$ | 9.789 | 0.002 | 9.782 | 0.002 | m s$^{-2}$ |
| $\langle\cos\theta\rangle$ | 0.999 | 0.002 | 0.875 | 0.100 | |
| $a^{\mathrm{eff}}$ | 9.779 | 0.020 | 8.56 | 0.98 | m s$^{-2}$ |

$h$, altitude; $s$, air speed; $a_x$, $a_y$, $a_z$, accelerations along $x$, $y$, $z$ axes of the vehicle; $\theta_x$, roll angle; $\theta_y$, slope angle; $\Omega_x$, $\Omega_y$, $\Omega_z$, rotation rates about the $x$, $y$, $z$ axes.
Values in the 'Mean' columns indicate the average of data recorded over five consecutive parabolas ($\sim$800 s of flight time), and the 'Range' column gives the interquartile range of the same data—indicating the typical variation for each parameter. The aircraft's altitude, air speed, roll and slope angles are courtesy of Novespace. The last three rows give the mean gravitational acceleration $\langle a\rangle$, the mean projection factor $\langle\cos\theta\rangle$ and the effective gravitational acceleration $a^{\mathrm{eff}}$ (equation (12)) used to measure the Eötvös parameter shown in Table 2. In these rows, the value in the Range column corresponds to the 1$\sigma$ uncertainty. Estimates of $\langle a\rangle$ were obtained from the Earth gravity model EGM2008 over the flight region defined by opposite-corner coordinates 6° 44′ W, 45° 23′ N and 2° 43′ W, 48° 37′ N at the indicated mean altitude $h$. The projection factor is based on the variation in the aircraft's roll and slope angles during the measurements.

coordinates at a fixed altitude $h$ using the Earth gravitational model EGM2008 (ref. 39). From these values, we calculate the average projection of the gravitational acceleration vector on the axis of the Raman beams ($\epsilon_k$). In the Earth frame, the interferometer axis is defined as $\epsilon_k = -\sin\theta_y\epsilon_{x'} + \sin\theta_x\cos\theta_y\epsilon_{y'} + \cos\theta_x\cos\theta_y\epsilon_{z'}$ after rotations about the $x'$ and $y'$ axes by roll angle $\theta_x$ and slope angle $\theta_y$, respectively. It follows that the effective gravitational acceleration is given by

$$a^{\mathrm{eff}} = \langle a(\varphi, \lambda, h)\rangle\langle\cos\theta_x\cos\theta_y\rangle, \tag{12}$$

where $\langle\cdots\rangle$ denotes an average. Table 2 contains the list of corrections applied to the raw data to obtain $\eta$. We now describe some of the dominant systematic effects that were specific to our experiment onboard the aircraft.

**Coriolis phase shift.** During steady flight, if the aircraft is tilted by angles $\theta_x$ and $\theta_y$ about the $x'$ and $y'$ axes (Fig. 1a), a component of the gravitational acceleration lies along the axes perpendicular to $\mathbf{k}^{\mathrm{eff}} = k^{\mathrm{eff}}\epsilon_z$, thus a rotation about these axes will cause a phase shift, $\phi^\Omega$, due to the Coriolis effect. To first order in the rotation rate $\mathbf{\Omega}$, this shift can be split into two main parts

$$\phi^\Omega = -2\left[\mathbf{k}^{\mathrm{eff}}\times(v_0 + a_0 T)\right]\cdot\mathbf{\Omega}T^2 = \phi_{v_0}^\Omega + \phi_{a_0}^\Omega, \tag{13}$$

where first term is due to an atomic velocity $\mathbf{v}_0$ at the start of the interferometer and the second originates from a constant acceleration $\mathbf{a}_0 = g\epsilon_{z'} + \delta\mathbf{a}$. For small angles

$\theta_x$ and $\theta_y$, it is straightforward to show that

$$\phi_{v_0}^\Omega = -2k^{\mathrm{eff}}\left(v_{0x}\Omega_y - v_{0y}\Omega_x\right)T^2, \tag{14}$$

$$\phi_{\delta a}^\Omega = -2k^{\mathrm{eff}}\left[(\delta a_x + g\theta_y)\Omega_y - (\delta a_y - g\theta_x)\Omega_x\right]T^3, \tag{15}$$

where $\delta\mathbf{a}$ is a small shot-to-shot variation due to the motion of the aircraft of order $|\delta\mathbf{a}| \simeq 0.05g$ (Fig. 4a), and the initial velocity is related to $\delta\mathbf{a}$ via $\mathbf{v}_0 = v_j^{\mathrm{sel}}\epsilon_z + \delta\mathbf{a}\Delta t$. Here $v_j^{\mathrm{sel}}$ is the selected atomic velocity determined by the frequency difference between Raman beams, and $\Delta t$ represents the free-fall time between cloud release and the first $\pi/2$-pulse ($\Delta t \simeq 3$ ms in our case).

Table 3 displays the mean value and range of variation of some inertial parameters during each flight configuration. These data imply that the dominant contribution to the Coriolis phase during steady flight is the instability in the roll angle. The corresponding phase shift at $T = 2$ ms is estimated to be $\phi_{1g}^\Omega \simeq 0.1(3)$ mrad for both $^{87}$Rb and $^{39}$K. In comparison, during a parabolic trajectory the atoms are in free-fall and the acceleration relative to the mirror is close to zero, hence the Coriolis phase shift is much less sensitive to the orientation of the aircraft relative to $\mathbf{g}$. However, during this phase the aircraft can reach rotation rates of $|\mathbf{\Omega}| > 5^\circ$ s$^{-1}$ (Fig. 4b), which occurs primarily about the $y$ axis (Fig. 1a). This causes small atomic velocities perpendicular to the direction of $\mathbf{k}^{\mathrm{eff}}$ to produce significant phase shifts. We estimate $\phi_{0g}^\Omega \simeq -3.7(3)$ mrad at $T = 2$ ms for a mean rotation rate of $\Omega_y \simeq 4.1^\circ$ s$^{-1}$.

These simple estimates, although useful to give an intuitive understanding, do not include effects due to finite Raman pulse lengths $\tau$, time-varying rotation rates $\mathbf{\Omega}(t)$ or time-varying accelerations $\mathbf{a}(t)$. Since these effects

are significant in our case, it was necessary to develop a new expression to accurately estimate the associated phase shift. The result of these calculations, which were based on the sensitivity function formalism[40], is the following expression

$$\Phi^\Omega = -\int w^\Omega(t)\left(\mathbf{k}^{\mathrm{eff}}\times\mathbf{v}(t)\right)\cdot\mathbf{\Omega}(t)\mathrm{d}t - \iint_t^\infty w^\Omega(t')\left(k^{\mathrm{eff}}\times a(t')\right)\cdot\mathbf{\Omega}(t)\mathrm{d}t'\mathrm{d}t,$$

(16)

which describes the Coriolis phase shift due to an atomic trajectory undergoing a time-dependent rotation $\mathbf{\Omega}(t)$ and acceleration $\mathbf{a}(t)$. Here $w^\Omega(t)$ is a weight function

$$w^\Omega(t) = tg^{\mathrm{s}}(t) + \int_t^\infty g^{\mathrm{s}}(t')\mathrm{d}t',$$

(17)

which contains the interferometer sensitivity function $g^{\mathrm{s}}(t)$. In the limit of short pulse lengths, and constant accelerations and rotations, equation (16) reduces to equation (13).

During the flight, we measure the acceleration of the Raman mirror in the rotating frame (Fig. 1) using a three-axis mechanical accelerometer, and the rotation rates $\Omega_x(t)$ and $\Omega_y(t)$ are measured using a two-axis fibre-optic gyroscope. The rotation rate about the $z$ axis of the aircraft, $\Omega_z$, does not contribute significantly to the Coriolis phase (since it is parallel to $\mathbf{k}^{\mathrm{eff}}$), hence precise measurements of this quantity using a third gyroscope were not required. We then integrate the equations of motion in the rotating frame to obtain the velocity $\mathbf{v}(t)$ relative to the Raman mirror, and we use equation (16) to obtain the Coriolis phase shift for each shot of the experiment. The values reported in the third row of Table 1 represent the average of this phase taken over the coarse of all measurements during a given flight configuration. For the DSD interferometer used in $0\,g$, we computed the Coriolis shift for both upward and downward atomic trajectories and combined the results as in equation (8).

**DSD phase shift.** The DSD interferometer that we use in microgravity is sensitive to an additional systematic shift that is not present in the single-diffraction interferometer. This phase shift arises from the fact that we cannot distinguish between the atoms that are diffracted upwards and downwards. For instance, if there is an asymmetry in the number of atoms diffracted along these two directions, and the direction-independent phase $\Sigma\Phi^{\mathrm{ind}}$ is non-zero, this will produce two phase-shifted fringe patterns with different contrasts. Since we measure the sum of these two fringe patterns, there is an additional phase shift that depends on the relative contrast $\varepsilon = C^-/C^+ - 1$ between the $\pm k^{\mathrm{eff}}$ interferometers and $\Sigma\Phi^{\mathrm{ind}}$ as follows

$$\phi^{\mathrm{DSD}} = \tan^{-1}\left[-\frac{\varepsilon/2}{1+\varepsilon/2}\tan\Sigma\Phi^{\mathrm{ind}}\right].$$

(18)

For the $T\simeq2\,\mathrm{ms}$ fringes shown in Fig. 3e, we estimate $\varepsilon\simeq0.05$ for both rubidium and potassium interferometers. Hence, using the total direction-independent systematics listed in columns 5 and 7 of Table 1, we obtain DSD phase shifts of $\phi_{\mathrm{Rb}}^{\mathrm{DSD}}\simeq-39(3)\,\mathrm{mrad}$ and $\phi_{\mathrm{K}}^{\mathrm{DSD}}\simeq-29(30)\,\mathrm{mrad}$.

**Quadratic Zeeman effect and magnetic gradient.** The primary source of systematic phase shift in this work originated from a time-varying $B$-field during the interferometer produced by a large aluminium breadboard near the coils used to produce a magnetic bias field for the interferometers. Owing to the relatively large pulsed fields ($\sim1.5\,\mathrm{G}$) required to sufficiently split the magnetically sensitive transitions in $^{39}\mathrm{K}$, Eddy currents produced in the aluminium breadboard during the interferometer significantly shift the resonance frequency of the clock transition $|1, m_{\mathrm{F}}=0\rangle\to|2, m_{\mathrm{F}}=0\rangle$ via the quadratic Zeeman effect. We recorded the field just outside the vacuum system with a flux gate magnetometer (Bartington MAG-03MCTPB500) and used these data, in conjunction with spectroscopic calibrations of the field at the location of the atoms, to compute the associated systematic phase shift for each shot of the experiment.

The second-order (quadratic) Zeeman effect shifts the frequency of the clock transition as $\Delta\omega_j^B = 2\pi K_j|B|^2$, where $K_{\mathrm{Rb}} = 575.15\,\mathrm{Hz\,G^{-2}}$ for $^{87}\mathrm{Rb}$ and $K_{\mathrm{K}} = 8513.75\,\mathrm{Hz\,G^{-2}}$ for $^{39}\mathrm{K}$ (ref. 41). This effect can shift the phase of the interferometers in three ways: (i) due to a $B$-field that is non-constant in time ($\phi_j^{B(t)}$); (ii) from a field that is non-constant in space ($\phi_j^{B(z)}$); or (iii) via the force on the atoms from a spatial magnetic gradient ($\phi_j^{\beta_1}$). The total systematic shift due to magnetic field effects is the sum of these three phases

$$\phi_j^{\mathrm{sys},B} = \phi_j^{B(t)} + \phi_j^{B(z)} + \phi_j^{\beta_1},$$

(19)

We model the local magnetic field experienced by the atoms as follows

$$B(z,t) = \beta_0\xi(t) + \beta_1 z$$

(20)

where $\beta_0$ is a magnetic bias field, $\beta_1 = \partial B/\partial z$ is a magnetic gradient and $\xi(t)$ is a unitless envelope function that can describe the field turn-on, as well as residual Eddy currents.

The phase shift due to a temporal variation of the $B$-field ($\phi_j^{B(t)}$) can be computed using[40]

$$\phi_j^{B(t)} = \int g_j^s(t)\Delta\omega_j^B(t)\mathrm{d}t = 2\pi K_j\int g_j^s(t)\left|B\left(z_j^0,t\right)\right|^2\mathrm{d}t,$$

(21)

where $g_j^s(t)$ is the interferometer sensitivity function[25,38] and $\Delta\omega_j^B(t) = 2\pi K_j|B(z_j^0,t)|^2$ is the clock shift at the initial position of the atoms. Similarly, the phase $\phi_j^{B(z)}$ due to the clock shift from a spatially non-uniform field can be expressed as

$$\phi_j^{B(z)} = 2\pi K_j\int g_j^s(t)\left(\left|B\left(\bar{z}_j(t),t\right)\right|^2 - \left|B\left(z_j^0,t\right)\right|^2\right)\mathrm{d}t,$$

(22)

Here $\bar{z}_j(t) = z_j^0 + \left(v_j^{\mathrm{sel}}\pm v_j^{\mathrm{rec}}/2\right)t + at^2/2$ is the centre-of-mass trajectory of atom $j$ along the interferometer pathways, $z_j^0$ and $v_j^{\mathrm{sel}}$ are the initial atomic position and selected velocity, respectively, $v_j^{\mathrm{rec}} = \hbar k_j^{\mathrm{eff}}/M_j$ is the corresponding recoil velocity and $a$ is a constant acceleration along the direction of $z$. We have ignored the influence of the magnetic gradient force on the atomic trajectory since it is small compared that of gravity. In equation (22), we have used the difference between the field experienced by a falling atom and that of a stationary atom at $z=z_j^0$ to separate the spatial effect of the field from the temporal one.

To measure $|B(z,t)|$, we used velocity-insensitive Raman spectroscopy of magnetically sensitive two-photon transitions ($|1, m_{\mathrm{F}}=\pm1\rangle\to|2, m_{\mathrm{F}}=\pm1\rangle$) and we extracted the resonance frequency as a function of the time in free fall in standard gravity—yielding a map of $|B(z,t)|$. However, this method cannot distinguish between the temporally and spatially varying components of the field. To isolate the spatial gradient $\beta_1$, we performed the same spectroscopy experiment with the bias field on continuously to eliminate the turn-on envelope and to minimize Eddy currents. The difference between these measurements yielded the temporally varying component of the field. For typical experimental parameters during the flight ($T_j\sim2\,\mathrm{ms}$, $\beta_0\sim1.5\,\mathrm{G}$), we find $\phi_{\mathrm{Rb}}^{B(t)}\sim2.1\,\mathrm{rad}$ and $\phi_{\mathrm{K}}^{B(t)}\sim30.5\,\mathrm{rad}$, as listed in the first row of Table 1. These relatively large phase shifts are produced by the large bias required to separate the $|1, m_{\mathrm{F}}=\pm1\rangle$ states from $|1, m_{\mathrm{F}}=0\rangle$, and a significant variation in the envelope during the interferometer (the field changes by $\sim0.5\,\mathrm{G}$ in 2 ms) produced by the Eddy currents. Similarly, we estimate $\phi_{\mathrm{Rb}}^{B(z)}\sim0.032\,\mathrm{rad}$ and $\phi_{\mathrm{K}}^{B(z)}\sim0.96\,\mathrm{rad}$ during $1\,g$, which arises from a measured gradient of $\beta_1\simeq13\,\mathrm{G\,m^{-1}}$, as listed in the second row of Table 1.

The phase shift $\phi_j^{\beta_1}$ arising from the force on the atoms due to the magnetic gradient can be computed by evaluating the state-dependent atomic trajectories and following the formalism of ref. 42. Up to order $T_j^4$ and $\Lambda_j^2 = hK_j/M_j$, this phase can be shown to be

$$\phi_j^{\beta_1} = \mp\frac{2}{3}k_j^{\mathrm{eff}}\left(\Lambda_j\beta_1\right)^2\left[\left(v_j^{\mathrm{sel}}\pm\frac{v_j^{\mathrm{rec}}}{2}\right)T_j + aT_j^2\right]T_j^2,$$

(23)

where the $\mp$ sign convention corresponds to $\pm k_j^{\mathrm{eff}}$. We emphasize that this phase scales $(\Lambda_{\mathrm{K}}/\Lambda_{\mathrm{Rb}})^2\sim33$ times more strongly for potassium than rubidium due to its lighter mass and smaller hyperfine splitting ($K_j\propto1/\omega_j^{\mathrm{HF}}$). However, since the magnetic gradient force is opposite in sign for $|F=1\rangle$ and $|F=2\rangle$, and the states are exchanged halfway through the interferometer, this phase shift is generally much smaller than those produced by shifts of the clock transition. For typical experimental parameters during steady flight ($T_j\simeq2\,\mathrm{ms}$, $v_j^{\mathrm{sel}}\simeq5\,\mathrm{cm\,s^{-1}}$, $a\simeq9.8\,\mathrm{m\,s^{-2}}$ and $\beta_1\simeq13\,\mathrm{G\,m^{-1}}$), the phase shift is $\phi_{\mathrm{K}}^{\beta_1}\simeq-0.12\,\mathrm{mrad}$ for $^{39}\mathrm{K}$ and $\phi_{\mathrm{Rb}}^{\beta_1}\simeq-3.4\,\mu\mathrm{rad}$ for $^{87}\mathrm{Rb}$. We sum this phase with $\phi_j^{B(z)}$ in the second row of Table 1.

**Data availability.** The authors declare that the primary data supporting the findings of this study are available within the article and its Supplementary Information file. Additional data are available from the corresponding author on request.

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

## Acknowledgements

This work is supported by the French national agencies CNES, ANR, DGA, IFRAF, action spécifique GRAM, RTRA 'Triangle de la Physique' and the European Space Agency. B. Barrett and L. Antoni-Micollier thank CNES and IOGS for financial support. P. Bouyer thanks Conseil Régional d'Aquitaine for the Excellence Chair. Finally, the ICE team thank A. Bertoldi of IOGS for his assistance during the Zero-G flight campaign in May 2015; V. Ménoret of MuQuans for helpful discussions; D. Holleville, B. Venon, F. Cornu of SYRTE and J.-P. Aoustin of the laboratory GEPI for their technical assistance building vacuum and optical components; and the staff of Novespace.

## Author contributions

P.B. and A.L. conceived the experiment and directed research progress; B. Battelier contributed to construction of the first-generation apparatus, helped to direct research progress and provided technical support; B. Barrett led upgrades to the second generation apparatus, performed experiments, carried out the data analysis and wrote the article; L.A.-M. and L.C. helped upgrade the potassium interferometer and carried out experiments; T.L. provided technical support during flight campaigns. All authors provided comments and feedback during the writing of this manuscript.

## Additional information

**Competing financial interests:** The authors declare no competing financial interests.

**Publisher's note**: 

