## [Peer Review File · Nature Communications]

Reviewers' comments:

Reviewer #1 (Remarks to the Author):

In their manuscript 'Dual Matter-Wave Inertial Sensors in Weightlessness' Brynle Barrett et al. describe an dual species atom interferometer experiment in a 'zero-g' environment. The experiment is the first to demonstrate a full stable atom interferometer in an zero-g environment and a full measurement done with the interferometer. They extract the 'Eötvös parameter' testing the equivalence principle from their data.

The experiment, even if its precision is not competitive with the best atom interferometer experiments, should be seen as a demonstrator of new enabling technology. Much was written about sending atom interferometers to space for precision measurements. This experiment is the first measurement in a zero-g environment, and to my knowledge also the first demonstration of an stable interferometer in zero-g (the others just showed fringes with random phase, and nothing could be measured).

Brynle Barrett et al. demonstrate a new interferometer that allows to compensate for the adverse effects during the parabolic flight. But even much more important the authors give a very detailed account of the adverse effects limiting the precision of their measurement, and relate them to adverse effects predicted in the proposed satellite experiments. As such the manuscript is of high interest and deserves publication in a high visibility journal.

The paper is well written, and includes a detailed analysis of the uncertainties and systematic errors. The data seems very robust and reliable. I recommend publication in nature communications.

I would like the authors to consider the following general remarks:

Why are these tests called 'quantum tests of free fall'? I know that this phrase even appears in titles of papers ... but what is tested is nothing more than a falling corner-cube will test. Gravity is assumed to be a classical potential, and one measures a Doppler shift due to the acceleration. The inertial mass and the gravitational mass enter in the same experiment. It is quantum mechanical superposition that is used to measure ... but does that make the experiment a 'quantum test of free fall'?

In the evaluation of the 'Eötvös parameter' testing the universality of free fall the authors discuss the systematic effects limiting their accuracy. I was surprised to see that the systematic effects were much larger in the normal measurement than in the 0-g measurement, resulting in a better measurement in zero-g than for $g=1$. How can one understand that, or was the 0-g measurement done with the compensated interferometer and the $g=1$ measurement with a regular interferometer? Why? If I understood correctly both measurements were done on the plane, but the $g=1$ measurements during regular flight.

In their measurement of the 'Eötvös parameter' testing the universality of free fall they use an interferometer that is compensated between the two species. At the level of the present accuracy of their measurement this is an easy task. Can the authors give an estimate how good the compensation needs to be for the envisioned precision tests at the 10^{-15} level?

Reviewer #2 (Remarks to the Author):

The authors demonstrate the first test of the equivalence principle with atoms in the weightless

environment produced during parabolic flight. The Eotvos parameter is measured by simultaneous ^{87}Rb - ^{39}K interferometers with systematic-limited uncertainties of $1.1\text{E-}3$ (1g) and $3.0\text{E-}4$ (0g). Although the values are far from the best precision achieved with atoms on Earth (^{87}Rb - ^{85}Rb $3.0\text{E-}8$, 1g), they do be the unique results under the most noisy and dynamic conditions (0.01 g/ $\sqrt{\text{Hz}}$, 0-1.8g, 5deg/s). The work represents a brave and necessary step toward the application of atom interferometers in future space and satellite missions.

The paper is good written. The treatment of uncertainties is suitable and previous works have been reasonably referred. I find this work novel and important. It deserves to be published in Nature Communications.

One question: To keep the scaling factor ratio $k=1$ the interrogation times T will not be the same for Rb and K interferometers. This time difference will inevitably bring in the coupling of noises to the Eotvos parameter measurement. How to estimate the contribution and how large the effect is in your typical experimental conditions?

Reviewer #3 (Remarks to the Author):

The authors present a technically challenging test of the universality of free fall (UFF) performed aboard an airplane with matter-wave inertial sensing. Limits on the Eotvos parameter were determined by using ^{39}K and ^{87}Rb as the test masses both in standard and microgravity conditions. This work represents the first quantum microgravity test of the UFF. The work follows earlier work performed by the same group that performed airborne standard and microgravity atom interferometry acceleration measurements for ^{87}Rb atoms, which was also published in Nature Communications (vol. 2, 474, 2011).

In addition to introducing the second species, this current work also introduces a clever way to perform interferometry at 0 g, which takes advantage of the velocity distribution of the atoms. This method may find use in space borne atom interferometry experiments that are still in the planning stage. The authors call the double single-diffraction (DSD). With a small Raman detuning from zero velocity, the method allows for simultaneous \pm keff measurements for each species, which allows for a high level of error and noise suppression. The work also presents a careful evaluation of all of the most significant systematic errors.

The experiment presented in the current manuscript is far more difficult than the previously published work and is definitely deserving of publication in Nature Communications. The manuscript is also well written and mostly free from typographical and grammatical errors. I recommend publication in Nature Communications after the authors address a few minor issues, listed below.

1. On page 3, it is implied but not explicitly stated that the two atom sources are co-located. I would explicitly state that they are. You could do this in the caption.
2. I think Figure 3 is a bit confusing. I would recommend that you state exactly what the interrogation times are for the two species to achieve the ellipses in Fig. 3f. As it is, you say that the interrogation times are approximately 1 and 2 ms, but that caused confusion for me since it is not possible to achieve $\kappa=1$ and have the same interrogation times for both species. I would be explicit.
3. On page 9 when you mention imaging the spatial fringes that result from rotation, you should consider citing a new paper that uses this method: Hoth et al., Applied Physics Lett., v. 109, 071113 (2016).
4. Line 201 under "methods": "...fixed to the rear of the retro-reflecting mirror."

5. Line 205 under "methods": "The chirp is disabled by software during parabolic maneuvers at 0 g."

6. Author Contributions: "B. Barret led upgrades..."

Reviewers' comments:

Reviewer #1 (Remarks to the Author):

In their manuscript 'Dual Matter-Wave Inertial Sensors in Weightlessness' Brynle Barrett et al. describe an dual species atom interferometer experiment in a 'zero-g' environment. The experiment is the first to demonstrate a full stable atom interferometer in an zero-g environment and a full measurement done with the interferometer. They extract the 'Eötvös parameter' testing the equivalence principle from their data.

The experiment, even if its precision is not competitive with the best atom interferometer experiments, should be seen as a demonstrator of new enabling technology. Much was written about sending atom interferometers to space for precision measurements. This experiment is the first measurement in a zero-g environment, and to my knowledge also the first demonstration of a stable interferometer in zero-g (the others just showed fringes with random phase, and nothing could be measured).

Brynle Barrett et al. demonstrate a new interferometer that allows to compensate for the adverse effects during the parabolic flight. But even much more important the authors give a very detailed account of the adverse effects limiting the precision of their measurement, and relate them to adverse effects predicted in the proposed satellite experiments. As such the manuscript is of high interest and deserves publication in a high visibility journal.

The paper is well written, and includes a detailed analysis of the uncertainties and systematic errors. The data seems very robust and reliable. I recommend publication in nature communications.

I would like the authors to consider the following general remarks:

Why are these tests called 'quantum tests of free fall'? I know that this phrase even appears in titles of papers ... but what is tested is nothing more than a falling corner-cube will test. Gravity is assumed to be a classical potential, and one measures a Doppler shift due to the acceleration. The inertial mass and the gravitational mass enter in the same experiment. It is quantum mechanical superposition that is used to measure ... but does that make the experiment a 'quantum test of free fall'?

RESPONSE: As in other articles in this field, here we used the term “quantum test” to point out that our test was carried out using microscopic sensors based on the quantum superposition principle, and to distinguish between other tests of the UFF that were done using macroscopic test masses. However, we agree with the reviewer that this terminology means nothing more than this, so we have replaced the two occurrences of “quantum test” with “test with quantum sensors” in the manuscript.

In the evaluation of the ‘Eötvös parameter’ testing the universality of free fall the authors discuss the systematic effects limiting their accuracy. I was surprised to see that the systematic effects were much larger in the normal measurement than in the 0-g measurement, resulting in a better measurement in zero-g than for $g=1$. How can one understand that, or was the 0-g measurement done with the compensated interferometer and the $g=1$ measurement with a regular interferometer? Why? If I understood correctly both measurements were done on the plane, but the $g=1$ measurements during regular flight.

RESPONSE: The level of systematic phase shifts were larger in 1g than in 0g because in 0g the interferometer configuration we used (the double single diffraction interferometer) was intrinsically insensitive to the largest systematic sources. During steady flight at 1g, we were forced to use the standard (single diffraction) interferometer due to the Doppler shift induced by the gravitational acceleration (the DSD interferometer requires a Doppler shift close to zero). As a result, the measurement in 1g reflected sensitivity to all systematic phase shifts present in the system.

We clarify this point on line 100 of the revised manuscript: “Fringes recorded in 1g were obtained with the single-diffraction interferometer along the $+k^{\text{eff}}$ direction, while those in 0g were realized using the DSD configuration along both $+/-k^{\text{eff}}$ simultaneously.”

In their measurement of the ‘Eötvös parameter’ testing the universality of free fall they use an interferometer that is compensated between the two species. At the level of the present accuracy of their measurement this is an easy task. Can the authors give an estimate how good the compensation needs to be for the envisioned precision tests at the 10^{-15} level?

RESPONSE: In our experiment, we compensate the scale factors so that their ratio (κ) is close to 1. There are a few reasons for this:

(1) As discussed in Ref. [26], we require $\kappa = 1$ in order to use parametric analysis methods like ellipse fitting. However, our correlative (FRAC) method does not rely on common mode rejection of vibration noise because we measure and correct for it. So even at the 10^{-15} level, this method does not require $\kappa = 1$.

(2) For experiments on ground, each atomic accelerometer measures a relatively large acceleration of $a = g = 9.8... \text{ m/s}^2$, hence the differential acceleration between test masses Δa (and hence the value of κ) needs to be known at the level of 10^{-15} . However, in free fall, the residual acceleration is much closer to zero than on ground. In this case, κ needs to be known only at the level of $10^{-15} * (g/\Delta a)$, which is a much less stringent requirement than on ground.

Reviewer #2 (Remarks to the Author):

The authors demonstrate the first test of the equivalence principle with atoms in the weightless environment produced during parabolic flight. The Eotvos parameter is measured by simultaneous ^{87}Rb - ^{39}K interferometers with systematic-limited uncertainties of $1.1\text{E-}3$ (1g) and $3.0\text{E-}4$ (0g). Although the values are far from the best precision achieved with atoms on Earth (^{87}Rb - ^{85}Rb $3.0\text{E-}8$, 1g), they do be the unique results under the most noisy and dynamic conditions (0.01 g/vHz, 0-1.8g, 5deg/s). The work represents a brave and necessary step toward the application of atom interferometers in future space and satellite missions.

The paper is good written. The treatment of uncertainties is suitable and previous works have been reasonably referred. I find this work novel and important. It deserves to be published in Nature Communications.

One question: To keep the scaling factor ratio $k=1$ the interrogation times T will not be the same for Rb and K interferometers. This time difference will inevitably bring in the coupling of noises to the Eotvos parameter measurement. How to estimate the contribution and how large the effect is in your typical experimental conditions?

RESPONSE: For the first part of this question, we refer to our response to question 3 of reviewer #1. For the second part, we note that our FRAC analysis technique (outlined in detail in Ref. [26]) is immune to vibration noise originating from the non-overlap of the interferometers. Hence, the contribution to final error on η due to this effect is negligible.

Reviewer #3 (Remarks to the Author):

The authors present a technically challenging test of the universality of free fall (UFF) performed aboard an airplane with matter-wave inertial sensing. Limits on the Eotvos parameter were determined by using ^{39}K and ^{87}Rb as the test masses both in standard and microgravity conditions. This work represents the first quantum microgravity test of the UFF. The work follows earlier work performed by the same group that performed airborne standard and microgravity atom interferometry acceleration measurements for ^{87}Rb atoms, which was also published in Nature Communications (vol. 2, 474, 2011).

In addition to introducing the second species, this current work also introduces a clever way to perform interferometry at 0 g, which takes advantage of the velocity distribution of the atoms. This method may find use in space borne atom interferometry experiments that are still in the planning stage. The authors call the double single-diffraction (DSD). With a small Raman detuning from zero velocity, the method allows for simultaneous +/- keff measurements for each species, which allows for a high level of error and noise suppression. The work also presents a careful evaluation of all of the most significant systematic errors.

The experiment presented in the current manuscript is far more difficult than the previously published work and is definitely deserving of publication in Nature Communications. The manuscript is also well written and mostly free from typographical and grammatical errors. I recommend publication in Nature Communications after the authors address a few minor issues, listed below.

1. On page 3, it is implied but not explicitly stated that the two atom sources are co-located. I would explicitly state that they are. You could do this in the caption.

RESPONSE: We agree with the reviewer. We have pointed this out in the caption of Figure 1 of the revised manuscript.

2. I think Figure 3 is a bit confusing. I would recommend that you state exactly what the interrogation times are for the two species to achieve the ellipses in Fig. 3f. As it is, you say that the interrogation times are approximately 1 and 2 ms, but that caused confusion for me since it is not possible to achieve $\kappa=1$ and have the same interrogation times for both species. I would be explicit.

RESPONSE: We agree with the reviewer. We have explained how κ was computed, along with the precise interrogation times and the pulse durations used for each species in the caption of Figure 3.

3. On page 9 when you mention imaging the spatial fringes that result from rotation, you should consider citing a new paper that uses this method: Hoth et al., Applied Physics Lett., v. 109, 071113 (2016).

RESPONSE: We have cited this article in the revised manuscript.

4. Line 201 under "methods": "...fixed to the rear of the retro-reflecting mirror."

RESPONSE: This typo has been corrected.

5. Line 205 under "methods": "The chirp is disabled by software during parabolic maneuvers at 0 g."

RESPONSE: This sentence has been corrected.

6. Author Contributions: "B. Barret led upgrades..."

RESPONSE: The typo has been corrected.

REVIEWERS' COMMENTS:

Reviewer #1 (Remarks to the Author):

The authors have fully accounted for all the comments. I recommend the paper to be published in Nature Communications.

many one small suggestion:

in their response the authors write:

During steady flight at 1g, we were forced to use the standard (single diffraction) interferometer due to the Doppler shift induced by the gravitational acceleration (the DSD interferometer requires a Doppler shift close to zero).

maybe this can also be mentioned in the text of the paper around line 100 where the comparison is discussed.

Reviewer #2 (Remarks to the Author):

No further question, the manuscript can be published as is.

Reviewer #3 (Remarks to the Author):

The authors have appropriately addressed all of the reviewers' comments, and I recommend publication in Nature Communications without further review. It is a very nice paper.

Reviewers' comments:

Reviewer #1 (Remarks to the Author):

The authors have fully accounted for all the comments. I recommend the paper to be published in Nature Communications.

many one small suggestion:

in their response the authors write:

During steady flight at 1g, we were forced to use the standard (single diffraction) interferometer due to the Doppler shift induced by the gravitational acceleration (the DSD interferometer requires a Doppler shift close to zero).

maybe this can also be mentioned in the text of the paper around line 100 where the comparison is discussed.

We agree with the referee's suggestion, and we have modified the sentence at line 100 to read:

“Due to the large Doppler shift induced by the gravitational acceleration, fringes recorded in $1g$ were obtained with the single-diffraction interferometer along the $+\mathbf{k}^{\text{eff}}$ direction. Matter-wave interference in $0g$ was realized using the DSD configuration along both $\pm\mathbf{k}^{\text{eff}}$ simultaneously, which requires a Doppler shift close to zero.”

Reviewer #2 (Remarks to the Author):

No further question, the manuscript can be published as is.

We thank the referee for their careful reading and praise of our manuscript.

Reviewer #3 (Remarks to the Author):

The authors have appropriately addressed all of the reviewers' comments, and I recommend publication in Nature Communications without further review. It is a very nice paper.

We thank the referee for their careful reading and praise of our manuscript.